# LONG-RANGE GRAPH WAVELET NETWORKS

## ABSTRACT

Modeling long-range interactions, the propagation of information across distant parts of a graph, is a central challenge in graph machine learning. Graph wavelets, inspired by multi-resolution signal processing, provide a principled way to capture both local and global propagation. However, existing wavelet-based graph neural networks rely on finite-order polynomial approximations, which limit their receptive fields and hinder long-range propagation. We propose Long-Range Graph Wavelet Networks (LR-GWN), which decompose wavelet filters into complementary local and global filters. Local aggregation is handled with efficient low-order polynomials, while long-range interactions are captured through a flexible spectral parameterization. This hybrid design unifies short- and long-distance information flow within a principled wavelet framework. Experiments show that LR-GWN achieves state-of-the-art performance among wavelet-based methods on long-range benchmarks, while remaining competitive on short-range datasets.

## 1 INTRODUCTION

Long-range interactions are central to complex systems, from electron correlations in quantum chemistry (Ambrosetti et al., 2014; Knörzer et al., 2022) to allosteric effects in biology (Dokholyan, 2016; Zhu et al., 2022). In graph neural networks (GNNs) (Gori et al., 2005; Scarselli et al., 2009; Gilmer et al., 2017; Bronstein et al., 2017), capturing these interactions requires models that can propagate information beyond local neighborhoods without the computational overhead of dense global interactions (Alon & Yahav, 2021; Dwivedi et al., 2022).

Wavelet-based graph neural networks (WGNNs) (Hammond et al., 2011), inspired by wavelet theory (Mallat, 1989), provide a principled framework for this challenge. In fact, WGNNs define spectral filters that can in principle capture different scales of information propagation through their filtering characteristics. However, designing exact wavelet filters face a fundamental computational bottleneck, as it would require computing the full eigenvalue decomposition of the graph Laplacian, which is prohibitive for large graphs.

To circumvent this cost, current WGNNs (Hammond et al., 2011; Xu et al., 2019; Liu et al., 2024a) approximate wavelet filters using low-order polynomials. While computationally efficient, these approaches suffer from two critical limitations. First, polynomial filters aggregate information only within a finite number of hops (Balcilar et al., 2020), inherently restricting their spatial reach. More fundamentally, polynomial approximations face an inherent trade-off between computational efficiency and functional expressiveness: while high-degree polynomials can theoretically approximate any function defined on the real interval as proved by the Weierstrass theorem, practically feasible orders cannot accurately represent the discontinuous or steep characteristics needed for selective frequency filtering, such as wavelets (Geisler et al., 2024).

Achieving both computational efficiency and sufficient functional expressiveness therefore requires rethinking how wavelet filters are parameterized. Existing polynomial-based approaches fundamentally fail to resolve this trade-off. On the one hand, low-order polynomials are highly efficient but their locality inherently confines propagation to a small neighborhood, preventing meaningful long-range interactions (Fig. 1a). On the other hand, increasing the polynomial order extends the receptive field and allows information to travel across more distant nodes, yet this comes at a rapidly growing computational cost and still does not provide true global coverage (Fig. 1b). Crucially, even very high polynomial degrees cannot faithfully reproduce the sharp spectral characteristics of wavelets, which are essential for selective frequency filtering. At the opposite end of the spectrum, an exact

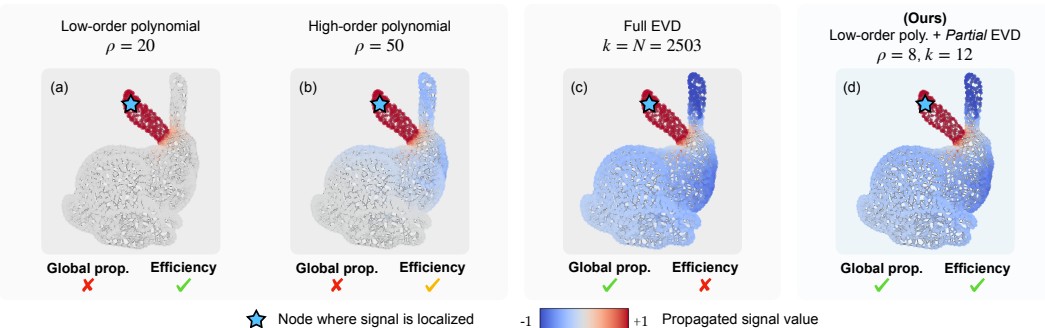

Figure 1: *Signal propagation under different Mexican hat wavelet filter approximations.* (a) Low-order polynomial ($\rho = 20$) restricts propagation to local neighborhoods. (b) Higher-order polynomial ($\rho = 50$) extends reach but remains spatially bounded. (c) Full EVD ($k = N = 2503$) enables global propagation at prohibitive cost. (d) LR-GWN ($\rho = 8, k = 12$) achieves global propagation with minimal computational overhead.

formulation based on a full eigenvalue decomposition of the graph Laplacian achieves principled and unrestricted global information flow. However, this approach scales cubically with the number of nodes and is thus infeasible for graphs of realistic size (Fig. 1c). These limitations highlight the need for a new design paradigm that bridges the gap between purely polynomial approximations, which are efficient but spectrally limited, and full spectral methods, which are expressive but computationally prohibitive.

We propose Long-Range Graph Wavelet Networks (LR-GWN), which overcome the limitations of polynomial-based WGNNs through a hybrid filter design. Our method decomposes each wavelet filter into two components: a low-order polynomial for efficient local aggregation and a spectral filter operating on the low-frequency eigenspace to enable long-range propagation (Fig. 1d). This approach requires only a partial eigendecomposition at preprocessing time, making it computationally practical while achieving the global reach that purely polynomial methods cannot provide. LR-GWN can operate under strict wavelet theory or with relaxed constraints for enhanced performance, providing a principled yet flexible framework for wavelet-based long-range graph learning.

**Our contributions** are as follows:

    I. **Hybrid parametrization** combining polynomial spatial filters with learnable spectral filters operating on truncated eigenspaces;

    II. **Principled wavelet framework** that maintains theoretical admissibility constraints while optionally allowing relaxation when needed;

    III. **Efficient implementation** requiring only partial eigenvalue decomposition with minimal preprocessing overhead, linear in the number of edges;

    IV. **State-of-the-art performance** among wavelet-based GNNs on long-range benchmarks, when both respecting wavelet admissibility criteria and relaxing them.

## 2 BACKGROUND

We recall the main tools from spectral graph theory and wavelet analysis; see Appendix A for a more detailed exposition.

**Graphs.** We consider undirected graphs $\mathcal{G} = (\boldsymbol{A}, \boldsymbol{X})$ with $n$ nodes, $m$ edges, an adjacency matrix $\boldsymbol{A} \in \{0, 1\}^{n \times n}$, and node features $\boldsymbol{X} \in \mathbb{R}^{n \times d}$. The degree matrix is $\boldsymbol{D} = \operatorname{diag}(\boldsymbol{A1})$, where $\boldsymbol{1}$ is the all-ones vector. From this, one can define the combinatorial Laplacian $\boldsymbol{L} = \boldsymbol{D} - \boldsymbol{A}$ and the symmetrically normalized Laplacian $\mathcal{L} = \boldsymbol{I} - \boldsymbol{D}^{-\frac{1}{2}} \boldsymbol{A} \boldsymbol{D}^{-\frac{1}{2}}$. In this work, we primarily use $\mathcal{L}$.

For undirected graphs, $\mathcal{L}$ is symmetric and positive semidefinite, which guarantees an eigendecomposition $\mathcal{L} = \boldsymbol{U} \boldsymbol{\Lambda} \boldsymbol{U}^{\top}$ with orthogonal eigenvectors $\boldsymbol{U}$ and eigenvalues $0 = \lambda_1 \leq \cdots \leq \lambda_n \leq 2$

(Chung, 1997). Small eigenvalues correspond to smooth, slowly varying components, while large eigenvalues capture oscillatory, high-frequency behavior.

**Graph Fourier Transform.** The eigenvectors of $\mathcal{L}$ form an orthogonal basis often referred to as the *graph Fourier basis*. Any graph signal $\boldsymbol{x} \in \mathbb{R}^n$ can be decomposed in this basis via the graph Fourier transform (GFT) as $\widehat{\boldsymbol{x}} = \boldsymbol{U}^\top \boldsymbol{x}$, with inverse $\boldsymbol{x} = \boldsymbol{U}\widehat{\boldsymbol{x}}$. Filtering with a spectral kernel $g : \mathbb{R} \to \mathbb{R}$ amounts to rescaling Fourier coefficients according to the eigenvalues:

$$g *_{\mathcal{G}} \boldsymbol{x} = \boldsymbol{U}\,\widehat{g}(\boldsymbol{\Lambda})\,\boldsymbol{U}^\top \boldsymbol{x},$$

where $\widehat{g}(\boldsymbol{\Lambda}) = \mathrm{diag}(\widehat{g}(\lambda_1), \dots, \widehat{g}(\lambda_n))$. In this sense, spectral filters are functions of the Laplacian spectrum, directly analogous to frequency-domain filters in classical Fourier analysis.

**Wavelets.** Wavelet theory extends Fourier analysis by enabling localized, multiresolution representations of signals (Mallat, 1989; 2009). A mother wavelet $\psi$ generates dilated and translated functions $\psi_{s,l}(t) = s^{-1/2}\psi\!\left(\frac{t-l}{s}\right)$, whose coefficients $W_f(s,l) = \int_{\mathbb{R}} f(t)\psi_{s,l}^*(t)\,dt$ decompose a signal across scales $s$ and locations $l$. Reconstruction requires the admissibility condition, $\widehat{\psi}(0) = 0$, ensuring $\widehat{\psi}$ acts as a band-pass filter. A complementary scaling function $\phi$ with $\widehat{\phi}(0) > 0$ provides low-pass coverage, yielding a multiresolution analysis where wavelets capture localized variations and the scaling function captures global trends.

**Wavelets on graphs.** The spectral graph wavelet transform (SGWT) (Hammond et al., 2011) adapts this construction to graphs. Given $\mathcal{L} = \boldsymbol{U}\boldsymbol{\Lambda}\boldsymbol{U}^\top$, a band-pass kernel $\widehat{\psi}$ defines the wavelet operator $\Psi = \boldsymbol{U}\,\widehat{\psi}(\boldsymbol{\Lambda})\,\boldsymbol{U}^\top$. Dilation to scale $s$ is achieved by $\widehat{\psi}(s\boldsymbol{\Lambda})$. A scaling function defines the low-pass operator $\Phi = \boldsymbol{U}\,\widehat{\phi}(\boldsymbol{\Lambda})\,\boldsymbol{U}^\top$. For a graph signal $\boldsymbol{x}$, the wavelet coefficients at scale $s$ are $W_{\boldsymbol{x}}(s) = \boldsymbol{U}\,\widehat{\psi}(s\boldsymbol{\Lambda})\boldsymbol{U}^\top \boldsymbol{x}$, while scaling coefficients are $H_{\boldsymbol{x}} = \boldsymbol{U}\,\widehat{\phi}(\boldsymbol{\Lambda})\boldsymbol{U}^\top \boldsymbol{x}$. Together, they provide a multiscale representation where $\widehat{\psi}$ captures localized variations (band-pass) and $\widehat{\phi}$ global structure (low-pass).

**Polynomial approximations.** Exact spectral filtering requires full eigendecomposition, which is impractical for large graphs. A common workaround is to approximate filters with low-order polynomials of $\mathcal{L}$, which can be applied directly in the vertex domain. Chebyshev expansions are particularly effective (Defferrard et al., 2016):

$$\widehat{g}_{\boldsymbol{\omega}}(\boldsymbol{\Lambda}) = \sum_{i=0}^{\rho} \omega_i T_i(\tilde{\boldsymbol{\Lambda}}),$$

where $\tilde{\boldsymbol{\Lambda}} = (2/\lambda_{\max})\boldsymbol{\Lambda} - \boldsymbol{I}$ rescales the spectrum to $[-1, 1]$.

**Definition 1** (Chebyshev Polynomials of the First Kind)**.** *The Chebyshev polynomials $T_i$ are defined recursively by*

$$T_0(x) = 1, \quad T_1(x) = x, \quad T_i(x) = 2xT_{i-1}(x) - T_{i-2}(x) \quad (i \geq 2).$$

Chebyshev filters require no eigendecomposition and correspond to message passing within at most $\rho$ hops on the graph (Balcilar et al., 2020), providing a scalable but inherently local approximation to spectral filters.

## 3 Limitations of Polynomial Graph Filters

Approximating spectral filters with polynomials avoids explicit eigendecomposition, since powers of $\mathcal{L}$ act directly in the vertex domain. For instance, a spectral filter of the form $\widehat{g}_{\boldsymbol{\omega}}(\boldsymbol{\Lambda}) = \sum_{i=0}^{\rho} \omega_i \boldsymbol{\Lambda}^i$ corresponds to the spatial operator $g_{\boldsymbol{\omega}}(\mathcal{L}) = \sum_{i=0}^{\rho} \omega_i \mathcal{L}^i$. This equivalence arises because powers of $\mathcal{L}$ in the vertex domain correspond exactly to powers of its eigenvalues in the spectral domain, and extends to other bases such as Chebyshev polynomials (see Appendix E.1). The main advantage of this parametrization lies in efficiency: recursive evaluation yields linear cost in the number of edges.

Despite these computational benefits, polynomial filters face structural barriers that limit their ability to capture long-range dependencies. We outline these issues from three complementary perspectives: (*i*) message propagation, (*ii*) function approximation, and (*iii*) convergence to discontinuous filters.

**Perspective I: Propagation.** Balcilar et al. (2020) showed that an $\ell$-layer message-passing GNN corresponds exactly to an $\ell$-order polynomial filter (i.e., with degree $\rho = \ell$). In practice, each layer adopts a first-order update of the form

$$\boldsymbol{H}^{(\ell)} = (\omega_0 \boldsymbol{I} + \omega_1 \mathcal{L}) \boldsymbol{H}^{(\ell-1)},$$

where multiplication by $\mathcal{L}$ aggregates information from one-hop neighbors (Kipf & Welling, 2016). Stacking $\ell$ such layers therefore propagates messages across at most $\ell$ hops. Beyond this range, information is inaccessible, and attempting to increase depth leads to *over-squashing* (Alon & Yahav, 2021), where exponentially many distant signals are compressed into fixed-size embeddings. Thus, polynomial filters cannot achieve effective long-range propagation without either prohibitive computational depth or architectural modifications.

> **Insight I:** Polynomial filters of degree $\rho$ propagate information at most $\rho$ hops, and increasing depth induces over-squashing, making them structurally unsuitable for long-range dependencies.

**Perspective II: Approximation.** From an approximation-theoretic viewpoint, low-degree polynomials are weak approximators for functions with steep transitions. This limitation is formalized by Markov's inequality:

**Theorem 1** (Markov's Inequality (Markov, 1889))**.** *Let $P : [-1, 1] \to [-1, 1]$ be a polynomial of degree at most $\rho$. Then*

$$\max_{x \in [-1,1]} \left| \frac{dP(x)}{dx} \right| \leq \rho^2.$$

This bound implies that the slope of a polynomial is fundamentally limited by $\rho^2$. While smooth functions can be represented with modest degree, approximating filters with sharp spectral cutoffs—such as scaling functions or low-scale wavelets—requires prohibitively large $\rho$. Empirically, medium-scale wavelets can be approximated relatively well by low-order polynomials, whereas scaling functions and

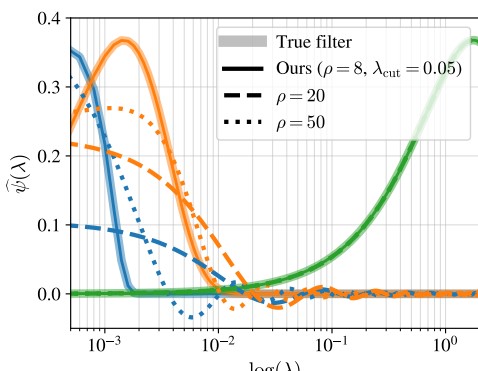

Figure 2: Different polynomial approx. vary in their ability to model low-frequency wavelets. Both low-order ($\rho = 20$) and high-order ($\rho = 50$) variants struggle with sharp transitions. Our method ($\rho = 8, \lambda_{\text{cut}} = 0.05$) captures both smooth and sharp irregularities.

fine-scale wavelets remain poorly represented unless the degree is dramatically increased (see Fig. 2). As a result, polynomial filters impose a strong smoothness bias that prevents them from capturing sharp spectral variations.

> **Insight II:** Approximating filters with steep transitions requires large polynomial degrees, making low-order polynomials unsuitable for modeling sharp spectral features.

**Perspective III: Convergence.** The most fundamental limitation arises for discontinuous spectral filters, which are common in graph learning (e.g., ideal band-pass kernels, step functions, or virtual-node mechanisms). While the Weierstrass theorem guarantees approximation for continuous functions, discontinuities fall outside its scope. Recently, Geisler et al. (2024) proved the following result:

**Theorem 2** (Slow Polynomial Convergence (Geisler et al., 2024))**.** *Let $\widehat{g}$ be a discontinuous spectral filter. For any polynomial approximation sequence $(g_{\gamma_\rho})_{\rho \in \mathbb{N}}$, there exists a graph sequence $(\mathcal{G}_\rho)_{\rho \in \mathbb{N}}$ such that*

$$\nexists \alpha > 0 : \sup_{0 \neq \boldsymbol{X} \in \mathbb{R}^{|\mathcal{G}_\rho| \times d}} \frac{\|(g_{\gamma_\rho} - g) *_{\mathcal{G}_\rho} \boldsymbol{X}\|_F}{\|\boldsymbol{X}\|_F} = \mathcal{O}(\rho^{-\alpha}).$$

In words, convergence to discontinuous filters is arbitrarily slow in operator norm, regardless of polynomial degree. This establishes a structural gap: polynomial filters are fundamentally incapable of efficiently approximating frequency-selective or discontinuous operators, even for large $\rho$.

**Insight III:** Polynomial filters converge extremely slowly to discontinuous spectral operators, making them fundamentally mismatched to long-range or frequency-selective behaviors.

**Summary:** Polynomial filters are efficient but inherently constrained. They are limited to $\rho$-hop propagation, biased toward smooth spectral responses, and unable to approximate discontinuous filters at any reasonable rate. These limitations highlight the necessity of alternative parametrizations that can reconcile efficiency with spectral expressiveness.

## 4 LONG-RANGE GRAPH WAVELET NETWORKS

We propose the *Long-Range Graph Wavelet Network (LR-GWN)*, a parametrization of graph wavelet filters that enables efficient, principled, and expressive filtering of graph signals, specifically tailored to capture long-range dependencies. Our design addresses three core limitations of prior approaches: it resolves the locality bottleneck of polynomial filters, preserves the interpretability of pure wavelet propagation, and achieves linear complexity with respect to the number of edges. The following subsections introduce its hybrid filter parametrization, wavelet-based propagation mechanism, and efficient implementation.

### 4.1 HYBRID PARAMETRIZATION OF WAVELET FILTERS

Effective graph filters must balance *efficiency*, *expressivity*, and *theoretical grounding*. Polynomial filters are attractive because they can be implemented recursively with linear cost in the number of edges. However, their inherent smoothness makes them poorly suited to approximate the sharp spectral transitions characteristic of wavelets. In contrast, purely spectral filters can realize principled wavelet propagation, but computing the full eigendecomposition of the Laplacian is prohibitive.

To reconcile these trade-offs, we introduce a *hybrid parametrization* that combines a polynomial backbone with a spectral correction. We recall that our goal is to parametrize wavelet filters, namely the scaling function $\widehat{\phi}$ and wavelet function $\widehat{\psi}$. At each layer $l$, both the wavelet kernel $\widehat{\psi}^{(l)}$ and the scaling function $\widehat{\phi}^{(l)}$ are modeled as the sum of local and global contributions:

$$\widehat{\psi}^{(l)}(\boldsymbol{\Lambda}) = P_{\boldsymbol{\omega}_\psi^{(l)}}(\boldsymbol{\Lambda}) + S_{\boldsymbol{\theta}_\psi^{(l)}}(\boldsymbol{\Lambda}), \qquad \widehat{\phi}^{(l)}(\boldsymbol{\Lambda}) = P_{\boldsymbol{\omega}_\phi^{(l)}}(\boldsymbol{\Lambda}) + S_{\boldsymbol{\theta}_\phi^{(l)}}(\boldsymbol{\Lambda}), \tag{1}$$

where $P_{\boldsymbol{\omega}^{(l)}}$ is a finite-order polynomial (spatial/local component) and $S_{\boldsymbol{\theta}^{(l)}}$ is a learnable spectral (global) component. The polynomial term efficiently captures localized interactions, while the spectral term affords precise control over selected frequency ranges, crucial for long-range propagation.

**Lemma 1.** *For a graph signal $\boldsymbol{x} \in \mathbb{R}^n$ and a filter kernel $\kappa$ parametrized as in Eq. (1), the filtering operation can be expressed as*

$$\kappa * \boldsymbol{x} = \boldsymbol{U}[P(\boldsymbol{\Lambda}) + S(\boldsymbol{\Lambda})]\boldsymbol{U}^\top \boldsymbol{x} = P(\mathcal{L})\boldsymbol{x} + \boldsymbol{U}S(\boldsymbol{\Lambda})\boldsymbol{U}^\top \boldsymbol{x}.$$

This decomposition highlights the dual nature of our filtering mechanism: the polynomial term $P(\mathcal{L})\boldsymbol{x}$ acts directly on $\mathcal{L}$ (i.e., the vertex domain), while the spectral correction $\boldsymbol{U}S(\boldsymbol{\Lambda})\boldsymbol{U}^\top \boldsymbol{x}$ acts on the eigenvalues (i.e., the spectral domain) and introduces frequency-selective adjustments. A proof is provided in Appendix E. Accordingly, at layer $l$ we define

$$\psi^{(l)}(\boldsymbol{U}, \boldsymbol{\Lambda}, \mathcal{L})\boldsymbol{x} = \big[P_{\boldsymbol{\omega}_\psi^{(l)}}(\mathcal{L}) + \boldsymbol{U}S_{\boldsymbol{\theta}_\psi^{(l)}}(\boldsymbol{\Lambda})\boldsymbol{U}^\top\big]\boldsymbol{x}, \tag{2}$$

$$\phi^{(l)}(\boldsymbol{U}, \boldsymbol{\Lambda}, \mathcal{L})\boldsymbol{x} = \big[P_{\boldsymbol{\omega}_\phi^{(l)}}(\mathcal{L}) + \boldsymbol{U}S_{\boldsymbol{\theta}_\phi^{(l)}}(\boldsymbol{\Lambda})\boldsymbol{U}^\top\big]\boldsymbol{x}. \tag{3}$$

**Truncated spectrum.** In practice, we compute a partial eigendecomposition $(\boldsymbol{U}, \boldsymbol{\Lambda}) = \text{EVD}(\mathcal{L}, k)$ of lowest $k$ eigenpairs with $k \ll n$, i.e., $\boldsymbol{U} \in \mathbb{R}^{n \times k}, \boldsymbol{\Lambda} \in \mathbb{R}^{k \times k}$. This focuses the spectral component on informative frequencies (e.g., low $\lambda$ for long-range effects) while maintaining scalability. The decomposition costs $\mathcal{O}(km)$, where $m \ll n^2$ is the number of edges, is computed once at preprocessing time and reused across layers.

**Filter parametrization.** We implement the spatial filter $P(\mathcal{L})$ using a Chebyshev basis for stable, recursive evaluation, and the spectral filter $S(\boldsymbol{\Lambda})$ via Gaussian smearing (Schütt et al., 2017), enabling fine-grained, band-specific control. Implementation details are given in Appendix B.1.

## 4.2 PRINCIPLED WAVELET PROPAGATION

A central feature of LR-GWN is that propagation is realized entirely through wavelet operators. Each layer applies a scaling function $\widehat{\phi}$ and a family of wavelet functions $\{\widehat{\psi}_j\}_{j=1}^{J}$, followed by pointwise nonlinearities and aggregation. The result is a propagation process that consists *solely* of wavelet-based transformations, preserving an interpretable, wavelet-based filtering operation.

Formally, given input features $\boldsymbol{H}^{(l-1)}$, one LR-GWN layer computes

$$\boldsymbol{H}^{(l)} = \sigma\Big[\phi^{(l)}(\boldsymbol{U}, \boldsymbol{\Lambda}, \mathcal{L})\, f_{\boldsymbol{\vartheta}}^{(l)}(\boldsymbol{H}^{(l-1)})\Big] \; \oplus \; \bigoplus_{j=1}^{J} \sigma\Big[\psi_j^{(l)}(\boldsymbol{U}, \boldsymbol{\Lambda}, \mathcal{L})\, f_{\boldsymbol{\vartheta}}^{(l)}(\boldsymbol{H}^{(l-1)})\Big], \qquad (4)$$

where $\sigma$ is a nonlinearity, $\oplus$ denotes aggregation, and $f_{\boldsymbol{\vartheta}}^{(l)}$ is a learnable feature transformation. In practice, we incorporate $f_{\boldsymbol{\vartheta}}^{(l)}(\boldsymbol{H}^{(l-1)})$ into each filter and compute, for the scaling function,

$$\phi^{(l)}(\boldsymbol{U}, \boldsymbol{\Lambda}, \mathcal{L})\, f_{\boldsymbol{\vartheta}}^{(l)}(\boldsymbol{H}^{(l-1)}) = \boldsymbol{U}\Big(S_{\boldsymbol{\theta}_{\phi}^{(l)}}(\boldsymbol{\Lambda}) \odot \big[\boldsymbol{U}^{\top} f_{\boldsymbol{\vartheta}}^{(l)}(\boldsymbol{H}^{(l-1)})\big]\Big) \;+\; P_{\boldsymbol{\omega}_{\phi}^{(l)}}(\mathcal{L})\, f_{\boldsymbol{\vartheta}}^{(l)}(\boldsymbol{H}^{(l-1)}) \quad (5)$$

with the wavelet functions defined analogously via $(\boldsymbol{\omega}_{\psi}^{(l)}, \boldsymbol{\theta}_{\psi}^{(l)})$.

We can ensure the band-pass behavior of the wavelet family, hence its wavelet theoretical validity, by enforcing the admissibility condition $\widehat{\psi}(0) = 0$, which guarantees invertibility of the wavelet transform. In practice, strict admissibility is not always necessary and relaxing it can yield slight empirical gains. A distinctive feature of LR-GWN is that it supports both regimes: admissibility can be enforced when theoretical guarantees are required, or relaxed when flexibility and performance are prioritized. We show how to enforce the constraint in Appendix B.2.2.

## 4.3 COMPUTATIONAL EFFICIENCY

LR-GWN achieves per layer linear complexity in the number of edges, $\mathcal{O}(m)$, by combining efficient polynomial filters ($\mathcal{O}(\rho m)$) with low-rank spectral projections ($\mathcal{O}(kdn)$). The key insight is that partial eigendecomposition with $k \ll n$ enables global spectral control at $\mathcal{O}(km)$ preprocessing cost, amortized across all layers. This contrasts with purely spectral methods requiring $\mathcal{O}(n^3)$ full EVD or polynomial methods limited to $\rho$-hop neighborhoods. Detailed analysis is provided in Appendix B.3.

**Summary.** LR-GWN offers a hybrid parametrization that unifies polynomial and spectral filters, a propagation scheme based solely on wavelet operators, and an implementation that scales linearly with the number of edges, as common GNNs. Together, these components provide an efficient, principled, and expressive framework for long-range graph signal filtering.

## 5 EMPIRICAL RESULTS

We evaluate LR-GWN on graph learning tasks requiring both long-range and short-range modeling capabilities. Our evaluation follows two axes: (*i*) comparing methods based on their adherence to pure wavelet propagation, distinguishing theory-compliant from theory-relaxed approaches, and (*ii*) demonstrating performance across interaction scales to validate the generality of our hybrid design.

**Experimental setup.** We conduct between 5 and 10 independent runs per experiment using different random seeds and report the mean performance and standard error of the mean. Our experiments are optimized for resource efficiency, requiring less than 10 GB of GPU memory, making them feasible on widely available hardware such as Nvidia GTX 1080Ti and 2080Ti GPUs. We use the AdamW optimizer (Loshchilov & Hutter, 2018) with cosine annealing scheduler (Loshchilov & Hutter, 2017) and linear warmup for stable training. Early stopping based on validation metrics prevents overfitting. All results use the *independent filter parametrization* (see Appendix B.1). We follow standard practice by augmenting node features with positional encodings (PEs) (Tönshoff et al., 2023), noting that our partial EVD provides PEs for free. Additional implementation details are provided in Appendix B.

**Wavelet methods categorization.** To fairly assess wavelet-based approaches, we distinguish between methods based on their theoretical adherence. *Theory-compliant* (TC) methods satisfy

Table 1: **Long-range benchmark results** by theoretical adherence. Theory-compliant methods (TC) satisfy wavelet admissibility and use pure wavelet propagation. Theory-relaxed methods (TR) either use additional non-wavelet modules, or relax theory constraints. Bold/underline represent first and second best, respectively.

| Method | PEPTIDES-FUNC (↑) | | PEPTIDES-STRUCT (↓) | |
|---|---|---|---|---|
| | TC | TR | TC | TR |
| SGWT (Hammond et al., 2011) | $60.23 \pm 0.27$ | - | $25.39 \pm 0.21$ | - |
| GWNN (Xu et al., 2019) | - | $65.47 \pm 0.48$ | - | $27.34 \pm 0.04$ |
| DEFT (Bastos et al., 2023) | - | $66.95 \pm 0.63$ | - | $25.06 \pm 0.13$ |
| WaveGC (Liu et al., 2024a) | - | $\underline{69.73 \pm 0.43}$ | - | $\underline{24.95 \pm 0.07}$ |
| **LR-GWN (ours)** | $\mathbf{70.52 \pm 0.29}$ | $\mathbf{72.16 \pm 0.41}$ | $\mathbf{24.63 \pm 0.07}$ | $\mathbf{24.62 \pm 0.06}$ |

admissibility constraints and propagate entirely through wavelet operations. Only SGWT (Hammond et al., 2011) and LR-GWN (with admissibility) qualify as theory-compliant. *Theory-relaxed* (TR) methods either incorporate auxiliary non-wavelet components (WaveGC (Liu et al., 2024a)) or relax theoretical constraints (LR-GWN (without admissibility), GWNN (Xu et al., 2019), DEFT (Bastos et al., 2023), ASWT-SGNN (Liu et al., 2024b)).

## 5.1 LONG-RANGE TASKS

**Datasets.** We evaluate on PEPTIDES-FUNC (multi-label classification of 10 functional classes) and PEPTIDES-STRUCT (node-level regression for 11 3D structural properties) from the Long-Range Graph Benchmark (Dwivedi et al., 2023; Tönshoff et al., 2023). These tasks specifically test long-range propagation in biological structures (peptides) where distant interactions determine physical conformations and biological functions.

**Quantitive Analysis.** LR-GWN achieves state-of-the-art performance on both datasets, as reported in Table 1. Most notably, it significantly outperforms all existing theory-compliant methods (70.52 vs 60.23 on PEPTIDES-FUNC) while maintaining theoretical guarantees. When constraints are relaxed, it surpasses all theory-relaxed baselines. It is worth noting that even the theory-compliant version of LR-GWN, achieves comparable performance to the theory-relaxed baselines, while maintaining pure wavelet interpretability and propagation.

**Training Time.** On PEPTIDES-FUNC, training LR-GWN takes approximately 20 seconds per epoch, totaling under 2.5 hours for full training (400 epochs). The partial EVD preprocessing costs roughly 1 minute, accounting for approximately 0.7% of total training time, highlighting that spectral augmentation via partial EVD remains computationally practical.

## 5.2 SHORT-RANGE TASKS

**Datasets.** We evaluate on PHOTO and COMPUTERS from Amazon co-purchase graphs (McAuley et al., 2015; Shchur et al., 2018). These datasets focus on local neighborhood interactions, validating that our long-range optimization does not compromise local modeling capabilities.

**Quantitive Analysis.** As reported in Table 2, LR-GWN achieves state-of-the-art performance on PHOTO and competitive results on COMPUTERS, demonstrating our proposed hybrid parametrization offers the flexibility to model both global and local propagation. Contrary to long-range tasks, relaxing admissibility constraints does not consistently improve performance on short-range datasets.

**Training Time.** On PHOTO, training LR-GWN takes approximately 3 sec. per epoch with total training time $\sim 20$ min. for the full run (400 epochs). The partial EVD preprocessing with $k = 50$ costs roughly 1.3 sec. using sparse solvers, accounting for approximately 0.1% of total training time.

Table 2: **Short-range benchmark results** by theoretical adherence. Theory-compliant methods (TC) satisfy wavelet admissibility and use pure wavelet propagation. Theory-relaxed methods (TR) either use additional non-wavelet modules, or relax theory constraints. Bold/underline represent first and second best, respectively.

| Method | PHOTO ($\uparrow$) | | COMPUTERS ($\uparrow$) | |
|---|---|---|---|---|
| | TC | TR | TC | TR |
| SGWT (Hammond et al., 2011) | $\underline{92.45 \pm 0.15}$ | - | $\underline{85.19 \pm 0.22}$ | - |
| GWNN (Xu et al., 2019) | - | $94.45 \pm 0.18$ | - | $90.75 \pm 0.16$ |
| ASWT-SGNN (Liu et al., 2025) | - | $93.80 \pm 0.12$ | - | $89.40 \pm 0.19$ |
| WaveGC (Liu et al., 2024a) | - | $\underline{95.37 \pm 0.44}$ | - | $\mathbf{92.26 \pm 0.18}$ |
| **LR-GWN (ours)** | $\mathbf{95.22 \pm 0.20}$ | $\mathbf{95.69 \pm 0.23}$ | $\mathbf{91.15 \pm 0.08}$ | $\underline{91.03 \pm 0.20}$ |

## 5.3 ABLATION STUDY

We assess the contribution of each component in LR-GWN for long-range tasks by performing an ablation study on PEPTIDES-FUNC.

**Component Analysis.** The spectral component proves essential for long-range modeling (3.34% performance drop when removed), while the polynomial component contributes primarily to computational efficiency with minimal performance impact (0.24% drop). This validates our design rationale: spectral filtering enables long-range propagation, while polynomial filtering provides efficient local aggregation.

Table 3: **Component ablation study on PEP-TIDES-FUNC.** We report absolute and relative drops in average precision ($\Delta_{AP}$) when removing individual components from LR-GWN.

| Ablation | $\Delta_{AP}$ (abs) | $\Delta_{AP}$ (%) |
|---|---|---|
| No Spatial Component | $-0.0017$ | $-0.24$ |
| No Spectral Component | $-0.0239$ | $-3.34$ |

**Admissibility Trade-off.** The performance difference between theory-compliant (70.52) and theory-relaxed (72.16) variants on PEPTIDES-FUNC demonstrates LR-GWN's flexibility in balancing theoretical guarantees with empirical performance, allowing adaptation to application-specific requirements.

**Considerations on $k$ and $\rho$.** We conducted preliminary ablation studies on both the number of retained eigenvalues $k$ and polynomial order $\rho$. While performance varies with both hyperparameters, no consistent pattern emerges across datasets. For smaller graphs like PEPTIDES-FUNC, we retain a large number of eigenvalues ($k = 150$) while keeping computational costs manageable, exploring the full expressive capacity. On larger graphs, smaller $k$ values typically suffice to balance efficiency and performance. Similarly, optimal $\rho$ is dataset-specific, but the model remains robust within reasonable ranges (typically $\rho = 3 - 15$).

We report additional results in Appendix D.

## 6 RELATED WORK

**Wavelets on Graphs.** The Spectral Graph Wavelet Transform (SGWT) by Hammond et al. (2011) introduced cubic spline wavelets and an exponential scaling function at fixed scales, ensuring admissibility but lacking learnable parameters. SGWT approximates filters using finite-order Chebyshev polynomials. Graph Wavelet Neural Networks (GWNN) (Xu et al., 2019) extend SGWT with fixed exponential wavelets, i.e., $g(s\lambda_i) = e^{s\lambda_i}$. While building on wavelet theory, this construction does not enforce the admissibility condition, since $g(0) = 1$. DEFT (Bastos et al., 2023) follows SGWT and parametrizes wavelet filters using free-parameters Chebyshev polynomials via a combination of GNNs and MLPs. The coefficients, however, are not bound and, by construction, do not enforce the admissibility condition. ASWT-SGNN (Liu et al., 2024b) approximates wavelet filter coefficients using Jackson–Chebyshev polynomials. Similarly to the discussion above about GWNN and DEFT, the coefficients are not bound to satisfy $g(0) = 0$. AGT (Cho et al., 2024) extends GWNN, and learns node-wise scales, either exactly or via approximation. As with the methods discussed so far, this

approach does not enforce the admissibility condition, and thus does not guarantee that the resulting filters behave as band-pass wavelets. WaveGC (Liu et al., 2025) satisfies the admissibility condition by parametrizing the scaling function with odd-terms of a Chebyshev polynomial, and the wavelet function with even-terms. However, the separation of basis terms limits functional expressiveness, and the use of non-wavelet-interpretable message passing diminishes the benefit of using wavelets.

**Hybrid Graph Neural Networks.** Hybrid GNNs combine local spatial and global spectral information to enhance long-range modeling. Stachenfeld et al. (2020) propose a hybrid message-passing framework that sacrifices permutation equivariance, while Liao et al. (2018) incorporate spectral information via the Lanczos algorithm. More recently, Geisler et al. (2024) introduced a spatial-spectral parametrization to improve efficiency. Beyond the explicit combination of spatial and spectral filters, graph rewiring (Gasteiger et al., 2019; Arnaiz-Rodríguez et al., 2022) enhance connectivity and mitigate over-squashing by modifying connectivity.

**Graph Transformers.** Graph Transformers (GTs) (Min et al., 2022; Dwivedi et al., 2021; Geisler et al., 2023) use global attention mechanisms to enable interactions between distant nodes. GTs often rely on spectral positional encodings paired with message passing (Rampášek et al., 2023; Geisler et al., 2023). Rampášek et al. (2023) introduce a framework for general, powerful and scalable graph transformers (GPS), which allow for the definition of a broad family of architectures that combine local message passing with global attention mechanisms. Similarly to these models, LR-GWN integrates local and global components. In our case, local aggregation is achieved through polynomial filters, while global propagation is handled via a spectral wavelet filter. However, the core methodology and motivation differ. LR-GWN is rooted in wavelet theory, with the aim of modeling low-pass and band-pass filters in a principled and interpretable manner. Graph transformers, by contrast, cover a broad family of architectures with varying attention mechanisms and are generally not designed with the same frequency-selective properties in mind. We report comparison to a well-established GT architecture that integrates local message passing with global attention, namely GPS (Rampášek et al., 2023), in Appendix D.

## 7 LIMITATIONS

While LR-GWN is effective and efficient in practice, it has a few inherent constraints. First, partial spectral decomposition, though more scalable than full EVD, can still be costly for very large graphs, especially when the number of retained eigenvalues $k$ increases. Second, the admissibility condition—used to ensure that wavelet filters behave as proper band-pass filters with desirable localization properties—provides useful theoretical grounding, but may limit flexibility in designing task-specific filters and might not be ideal in all scenarios. Lastly, our current formulation emphasizes low-frequency components by selecting the smallest $k$ eigenvalues and corresponding eigenvectors; however, this is not a fundamental limitation—LR-GWN can be extended to focus on other spectral regions, such as mid- or high-frequency bands, by selecting different eigenvalue ranges.

## 8 DISCUSSION AND CONCLUSION

We revisited the limitations of wavelet GNNs constrained by polynomial approximations and introduced Long-Range Graph Wavelet Networks (LR-GWN), a hybrid design that couples polynomial aggregation with spectral corrections on truncated eigenspaces. This construction unifies local and global information flow within wavelet operators, while supporting both strict admissibility for theoretical guarantees and relaxed variants for empirical gains.

Empirically, LR-GWN achieves state-of-the-art performance across wavelet-based models on long-range benchmarks and remains competitive on short-range tasks. More importantly, LR-GWN establishes wavelet-based filters as interpretable and theoretically grounded tools for graph learning, laying the foundation for adaptive and more expressive architectures. With LR-GWN, we aim to advance both the theoretical foundations and practical applications of wavelet-based long-range graph representation learning.

## REPRODUCIBILITY STATEMENT

We have taken several steps to ensure the reproducibility of our results. A complete description of the mathematical foundations, including background on spectral graph theory, wavelet analysis, and polynomial approximations, is provided in Section 2 and Appendix A. The architecture of LR-GWN, including its hybrid parametrization, wavelet propagation, and computational complexity, is detailed in Section 4 of the main paper and further elaborated in Appendix B (implementation details, admissibility condition, and residuals). Hyperparameters, training setup, and dataset descriptions are specified in Section 5 and Appendix C, with ablation studies reported in Section 5.3 and Appendix D. Proofs of theoretical claims (e.g., Lemma 1 and the equivalence of polynomial filters in spectral and spatial domains) are provided in Appendix E. For empirical reproducibility, we conduct multiple independent runs per experiment and report means and unbiased standard errors (Section 5). All experiments are based on the standardized LRGB benchmark framework (Dwivedi et al., 2022; Tönshoff et al., 2023).

## USE OF LARGE LANGUAGE MODELS

We used large language models (LLMs) solely for polishing the writing of the paper. The models were applied to improve clarity, grammar, and style in parts of the text. No LLMs were used for generating research ideas, designing experiments, analyzing results, or drawing conclusions. All scientific content and contributions are the responsibility of the authors.

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

## APPENDIX

This appendix provides additional technical material, implementation details, and extended results that complement the main text. We first expand the theoretical and mathematical background underlying our approach, including graphs and Laplacians, the graph Fourier transform, wavelet theory, and polynomial approximations. We then provide additional implementation details (filter parametrization, admissibility condition, residuals, and complexity analysis), and extended empirical results such as ablation studies and comparisons against broader baselines. Finally, we include complete proofs of the theoretical statements referenced in the main paper.

## A EXTENDED BACKGROUND

This appendix provides a more detailed overview of the mathematical foundations underlying our method. We first recall core concepts from spectral graph theory, then review wavelet analysis in both Euclidean and graph domains, and finally introduce polynomial approximations that enable scalable implementations.

### A.1 GRAPHS AND LAPLACIANS

A graph $\mathcal{G} = (\boldsymbol{A}, \boldsymbol{X})$ consists of $n$ nodes, $m$ edges, an adjacency matrix $\boldsymbol{A} \in \{0, 1\}^{n \times n}$, and node features $\boldsymbol{X} \in \mathbb{R}^{n \times d}$. The degree matrix is defined as $\boldsymbol{D} := \mathrm{diag}(\boldsymbol{A1}) \in \mathbb{R}^{n \times n}$, where $\boldsymbol{1} \in \mathbb{R}^n$ denotes the all-ones vector. From this, the combinatorial Laplacian is $\boldsymbol{L} := \boldsymbol{D} - \boldsymbol{A}$, while the symmetrically normalized Laplacian is $\mathcal{L} := \boldsymbol{I} - \boldsymbol{D}^{-\frac{1}{2}} \boldsymbol{A} \boldsymbol{D}^{-\frac{1}{2}}$. In this work we primarily use $\mathcal{L}$, though other variants (e.g., random-walk Laplacian) are also common.

For undirected graphs, $\mathcal{L}$ is symmetric and positive semidefinite, which guarantees an eigenvalue decomposition $\mathcal{L} = \boldsymbol{U}\boldsymbol{\Lambda}\boldsymbol{U}^\top$. Here $\boldsymbol{U} \in \mathbb{R}^{n \times n}$ contains the eigenvectors as columns, forming an orthogonal basis for functions defined on the graph ($\boldsymbol{U}\boldsymbol{U}^\top = \boldsymbol{I}$), and $\boldsymbol{\Lambda} = \mathrm{diag}(\lambda_1, \ldots, \lambda_n)$ collects the eigenvalues. The spectrum is bounded as $0 = \lambda_1 \leq \cdots \leq \lambda_n \leq 2$. Small eigenvalues correspond to smooth, slowly varying components, while large eigenvalues capture oscillatory, high-frequency behavior (Chung, 1997).

### A.2 GRAPH FOURIER TRANSFORM

The graph Fourier transform (GFT) (Shuman et al., 2011; Sandryhaila & Moura, 2013) generalizes the classical discrete Fourier transform (DFT) (Bracewell, 2000) to signals supported on graphs. In the classical Euclidean setting, a signal is decomposed into complex exponentials, whose frequencies correspond to sinusoidal oscillations of different scales. On a graph, there is no notion of translation

or frequency in the usual sense. Instead, the eigenvectors of the Laplacian play the role of Fourier modes, with the associated eigenvalues serving as the notion of frequencies.

Given a graph signal $\boldsymbol{x} \in \mathbb{R}^n$ (often corresponding to a column of the feature matrix $\boldsymbol{X}$), the GFT is $\widehat{\boldsymbol{x}} = \boldsymbol{U}^\top \boldsymbol{x}$, which projects the signal onto the eigenbasis of $\mathcal{L}$. The inverse GFT reconstructs the signal as $\boldsymbol{x} = \boldsymbol{U}\widehat{\boldsymbol{x}}$. Filtering with a spectral kernel $g : \mathbb{R} \to \mathbb{R}$ amounts to rescaling each Fourier coefficient according to the eigenvalue, yielding

$$g *_{\mathcal{G}} \boldsymbol{x} = \boldsymbol{U}\, \widehat{g}(\boldsymbol{\Lambda})\, \boldsymbol{U}^\top \boldsymbol{x},$$

where $\widehat{g}(\boldsymbol{\Lambda}) = \mathrm{diag}(\widehat{g}(\lambda_1), \ldots, \widehat{g}(\lambda_n))$, and $\widehat{g} = \boldsymbol{U}^\top g$. In this sense, spectral filters are functions of the Laplacian spectrum, directly analogous to frequency filters in classical Fourier analysis.

### A.3 WAVELET ANALYSIS

Wavelet theory extends Fourier analysis by enabling localized, multiresolution representations of signals (Mallat, 1989; 2009). Whereas Fourier bases are global and capture only frequency information, wavelets provide both spectral and spatial localization by combining scaling and translation operations.

The continuous wavelet transform (CWT) of a function $f : \mathbb{R} \to \mathbb{R}$ at scale $s > 0$ and location $l \in \mathbb{R}$ is defined as

$$W_f(s, l) = \int_{\mathbb{R}} f(t)\, \psi_{s,l}^*(t)\, dt,$$

where the wavelet family is generated from a mother wavelet $\psi$ by scaling and translation, $\psi_{s,l}(t) = \frac{1}{\sqrt{s}} \psi\!\left(\frac{t-l}{s}\right)$, and $^*$ denotes complex conjugation. Exact reconstruction requires the mother wavelet to satisfy the admissibility condition, ensuring that no frequency component is lost.

**Proposition 1** (Wavelet Admissibility). *A wavelet $\psi$ with Fourier transform $\widehat{\psi}$ is admissible if*

$$C_\psi = \int_0^\infty \frac{|\widehat{\psi}(\zeta)|^2}{\zeta}\, d\zeta < \infty,$$

*which holds when $\widehat{\psi}(0) = 0$ and $\lim_{\zeta \to \infty} \widehat{\psi}(\zeta) = 0$.*

This condition ensures that $\widehat{\psi}$ behaves as a band-pass filter, emphasizing intermediate frequency bands while attenuating very low and high frequencies. To retain the low-frequency components, wavelet systems are typically complemented by a scaling function $\phi$ with Fourier transform $\widehat{\phi}(0) > 0$, which acts as a low-pass filter. Together, $\phi$ and the wavelet family provide a multiresolution decomposition of signals.

### A.4 WAVELETS ON GRAPHS

Hammond et al. (2011) extend wavelet analysis to graphs by defining filters in the Laplacian spectral domain. Let $\mathcal{L} = \boldsymbol{U}\boldsymbol{\Lambda}\boldsymbol{U}^\top$ be the eigendecomposition of the (normalized) Laplacian, and let $\widehat{\psi}$ and $\widehat{\phi}$ denote a band-pass wavelet kernel and a low-pass scaling kernel, respectively (both applied elementwise to eigenvalues).

The *wavelet operator* at scale $s > 0$ and the *scaling operator* are

$$\Psi_s := \boldsymbol{U}\, \widehat{\psi}(s\boldsymbol{\Lambda})\, \boldsymbol{U}^\top, \qquad \Phi := \boldsymbol{U}\, \widehat{\phi}(\boldsymbol{\Lambda})\, \boldsymbol{U}^\top.$$

Given a graph signal $\boldsymbol{x} \in \mathbb{R}^n$, the wavelet coefficients at scale $s$ are $W_{\boldsymbol{x}}(s) = \Psi_s \boldsymbol{x}$, while the scaling coefficients are $H_{\boldsymbol{x}} = \Phi \boldsymbol{x}$. This parallels the classical construction: wavelets capture localized variations, whereas the scaling function captures smooth, global structure. A multiscale analysis is obtained by a family of dilations $\{s_j\}$, i.e., $\{\widehat{\psi}(s_j \cdot)\}_j$. On graphs, admissibility is typically enforced by $\widehat{\psi}(0) = 0$, with low frequencies covered by $\widehat{\phi}$ satisfying $\widehat{\phi}(0) > 0$.

### A.5 CHEBYSHEV POLYNOMIAL APPROXIMATIONS

Although spectral constructions are mathematically elegant, direct computation is expensive due to the eigenvalue decomposition, which scales cubically in the number of nodes. To make spectral filters

scalable, they are commonly approximated with low-order polynomials in the Laplacian, which can be applied directly in the vertex domain.

A widely used choice is the Chebyshev polynomial basis (Hammond et al., 2011; Defferrard et al., 2016). For a filter $\widehat{g}$, one approximates

$$\widehat{g}_{\boldsymbol{\omega}}(\boldsymbol{\Lambda}) = \sum_{i=0}^{\rho} \omega_i T_i(\tilde{\boldsymbol{\Lambda}}),$$

where $\boldsymbol{\omega} = [\omega_0, \dots, \omega_\rho]$ are learnable coefficients, $\rho$ is the polynomial order, and $\tilde{\boldsymbol{\Lambda}} = \frac{2}{\lambda_{\max}}\boldsymbol{\Lambda} - \boldsymbol{I}$ rescales the spectrum to $[-1, 1]$, the domain of the Chebyshev basis.

**Definition 2** (Chebyshev Polynomials of the First Kind). *The Chebyshev polynomials $T_i$ are defined recursively by*

$$T_0(x) = 1, \quad T_1(x) = x, \quad T_i(x) = 2xT_{i-1}(x) - T_{i-2}(x) \quad (i \geq 2).$$

Chebyshev expansions are particularly effective because they minimize the maximum approximation error on $[-1, 1]$ (in the minimax sense), and can be evaluated efficiently via recurrence. In practice, this enables fast, localized, and scalable approximations of spectral filters, including graph wavelets, without explicit eigendecomposition.

# B IMPLEMENTATION DETAILS

## B.1 FILTER PARAMETRIZATION

**Spatial Filter.** We model the spatial filter $P_{\boldsymbol{\omega}^{(l)}} : [0, \lambda_{\max}]^k \to \mathbb{R}^k$ using a finite-order Chebyshev polynomial expansion with learnable coefficients $\boldsymbol{\omega}^{(l)} = [\omega_0^{(l)}, \dots, \omega_\rho^{(l)}] \in \mathbb{R}^{\rho+1}$:

$$P_{\boldsymbol{\omega}^{(l)}}(\mathcal{L}) = \sum_{i=0}^{\rho} \omega_i^{(l)} T_i(\mathcal{L}), \tag{6}$$

where $T_i(\cdot)$ is the $i$-th Chebyshev polynomial. This formulation enables spatially localized filtering with computational efficiency. Alternative polynomial bases, such as Bernstein (He et al., 2021) or Jacobi polynomials (Wang & Zhang, 2022), could also be employed.

**Spectral Filter.** The spectral component $S_{\boldsymbol{\theta}^{(l)}} : [0, \lambda_{\max}]^k \to \mathbb{R}^k$ operates directly on the eigenvalues of $\mathcal{L}$, offering fine-grained frequency control. It is defined as

$$S_{\boldsymbol{\theta}^{(l)}}(\lambda) = \mathrm{GaussianSmearing}(\lambda)\boldsymbol{W}_{\theta^{(l)}}, \tag{7}$$

where $\boldsymbol{W}_{\theta^{(l)}} \in \mathbb{R}^{z \times d^{(l)}}$ are learnable weights, and $d^{(l)}$ is the output dimensionality of layer $l$. The basis $\mathrm{GaussianSmearing}(\lambda) : [0, 2]^k \to \mathbb{R}^{k \times z}$ projects eigenvalues onto $z$ Gaussian radial functions, introducing a frequency cutoff $\lambda_{\mathrm{cut}} < \lambda_{\max}$ (Schütt et al., 2017). We follow Geisler et al. (2024) and apply a frequency-domain windowing function to regularize the spectral response. Element-wise, the filter is defined as $S_{\boldsymbol{\theta}^{(l)}}(\boldsymbol{\Lambda})_{u,v} := S_v(\lambda_u; \boldsymbol{\theta}^{(l)})$. Alternative spectral parametrizations, such as SpecFormer (Bo et al., 2023), could be used depending on application-specific needs.

### B.1.1 SHARED AND INDEPENDENT FILTER CONFIGURATIONS

We define two configurations for our wavelet-based graph neural network: the *shared filter* and the *independent filter* approaches. In the shared filter configuration, we parametrize a single mother wavelet and generate a family of self-similar wavelet functions by applying predefined or learnable scales. Conversely, in the independent filter approach, each wavelet function is parametrized independently, allowing greater flexibility and expressivity.

**Shared Filter.** In this configuration, the $j$-th wavelet filter is constructed as

$$\psi\left(\boldsymbol{U}, \boldsymbol{\Lambda}, \mathcal{L}; s_j\right) = \boldsymbol{U}S_{\theta_\psi}(s_j\boldsymbol{\Lambda})\boldsymbol{U}^\top + P_{\omega_\psi}(s_j\mathcal{L}), \tag{8}$$

where a single function $\psi$ is parametrized following Eq. (3) and then scaled using predefined or learnable scales (Appendix B.1.2). This method is computationally efficient, as it requires only a single set of parameters $\theta_\psi$ and $\omega_\psi$, reducing the overall complexity of the model.

**Independent Filters.**   To increase model expressivity, we introduce an alternative approach in which each wavelet function is parametrized independently instead of being derived from a single mother wavelet. This configuration allows each wavelet to adjust its spectral and polynomial components separately, adapting to different levels of smoothness or sharpness. Larger-scale wavelets tend to be smoother and benefit more from polynomial components, while smaller-scale wavelets are sharper and require a stronger contribution from the spectral component. In the independent filters configuration, the $j$-th wavelet filter is given by

$$\psi_j\left(\boldsymbol{U}, \boldsymbol{\Lambda}, \mathcal{L}\right) = \boldsymbol{U} S_{\theta_{\psi_j}}(\boldsymbol{\Lambda})\boldsymbol{U}^\top + P_{\omega_{\psi_j}}(\mathcal{L}), \tag{9}$$

where each $\psi_j$ has its own set of learnable spectral parameters $\theta_{\psi_j}$ and polynomial parameters $\omega_{\psi_j}$.

### B.1.2   SCALE PARAMETRIZATION

Wavelet-based graph filters rely on a set of scales $\{s_j\}_{j=1}^J$ to capture multi-resolution information. Instead of using fixed scales as in Hammond et al. (2011), we introduce a learnable interpolation scheme in the shared filter configuration (Appendix B.1.1). The scale bounds are defined as $\log s_{\min} = \log(t_{\mathrm{L}}/\lambda_{\max})$ and $\log s_{\max} = \log(t_{\mathrm{U}}\lambda_{\mathrm{LP}}/\lambda_{\max})$, where $t_{\mathrm{L}}$ and $t_{\mathrm{U}}$ are learnable lower and upper limits, and $\lambda_{\mathrm{LP}}$ emphasizes low-frequency components. Assuming the maximum eigenvalue of the graph to be $\lambda_{\max} = 2$ as it is usually the case in real-world graphs, we have $\log s_{\min} = \log t_{\mathrm{L}}, \log s_{\max} = \log t_{\mathrm{U}}\lambda_{\mathrm{LP}}$. The intermediate scales are uniformly spaced in log-space,

$$s_i = \exp\left(\log s_{\max} + \frac{i(\log s_{\min} - \log s_{\max})}{N-1}\right). \tag{10}$$

To ensure positive scales, we apply the softplus function.

### B.2   MODELING PRACTICES

### B.2.1   INITIALIZATION

The wavelet filter parameters are initialized using standard Xavier initialization (Glorot & Bengio, 2010) for neural network weights and by setting $\omega_0 = 1$ and $\omega_i = 0$ for all $i = 1, \ldots, \rho$ in the Chebyshev polynomials. While the spectral and spatial components could theoretically be designed to match a predefined wavelet, such as the Mexican hat wavelet, we observed no significant improvements from this approach. Consequently, we adopt conventional parameter initialization without enforcing wavelet-based constraints in our experiments.

### B.2.2   ADMISSIBILITY CONDITION

To ensure the theoretical validity of our graph wavelet filter, we must guarantee the invertibility of the wavelet transform. This requires satisfying the admissibility condition at zero frequency, as stated in Appendix 1. Since the graph spectrum is bounded (i.e., $\lambda \in [0, \lambda_{\max}]$ with $\lambda_{\max} \leq 2$), we only need to enforce the condition $\hat{\psi}(0) = 0$, while disregarding $\lim_{\lambda \to \infty} \hat{\psi}(\lambda) = 0$.

Enforcing the admissibility condition is not always a necessity. Rather, it is a theoretical property that ensures the wavelet behaves as a band-pass filter. In practice, many applications do not strictly require this condition, and relaxing it can lead to slight empirical gains in some cases. However, with LR-GWN, we aim to provide a theoretically grounded framework that can allow to enforce the admissibility condition if needed, while also permitting for flexibility in applications where this condition is not strictly required.

**Primary Approach.**   In practice, we can enforce this constraint by subtracting the zero-frequency response from the spectral and polynomial filters:

$$\widetilde{P}_\omega(\lambda) = P_\omega(\lambda) - P_\omega(0), \quad \widetilde{S}_\theta(\lambda) = S_\theta(\lambda) - S_\theta(0). \tag{11}$$

This transformation ensures $\widehat{\psi}(0) = \widetilde{P}_\omega(0) + \widetilde{S}_\theta(0) = 0$ by construction. Thus, the wavelet filter in the spectral domain becomes

$$\widehat{\psi}(\boldsymbol{\Lambda}) = \widetilde{P}_\omega(\boldsymbol{\Lambda}), + \widetilde{S}_\theta(\boldsymbol{\Lambda}) \tag{12}$$

and the filtering operation in the vertex domain is given by

$$\psi\left(\boldsymbol{U}, \boldsymbol{\Lambda}, \mathcal{L}\right) = \widetilde{P}_\omega(\mathcal{L}), + \boldsymbol{U}\widetilde{S}_\theta(\boldsymbol{\Lambda})\boldsymbol{U}^\top \tag{13}$$

where both $\widetilde{P}_\omega(\mathcal{L})$ and $\widetilde{S}_\theta(\boldsymbol{\Lambda})$ are zero-frequency corrected versions of their respective filters.

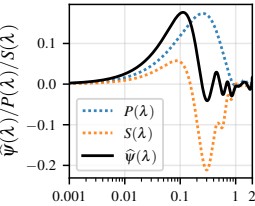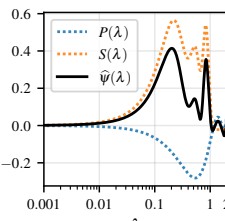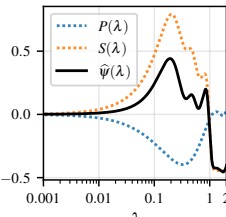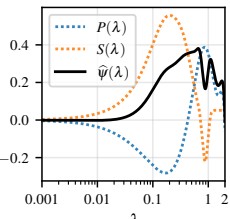

Figure 3: Examples of filters learned by our model during training. The polynomial component $P(\lambda)$ captures the smooth part of the filter, while the spectral component $S(\lambda)$ introduces flexibility, enabling sharper variations. Our filters inherently satisfy the admissibility condition $\widehat{\psi}(0) = 0$.

**Alternative Approach.** For the wavelet filter $\psi(\mathbf{\Lambda}) = S_\theta(\mathbf{\Lambda}) + P_\omega(\mathbf{\Lambda})$, where $P_\omega(\mathbf{\Lambda})$ is expressed as a Chebyshev polynomial, the admissibility condition at zero frequency is satisfied if

$$\omega_0 = \sum_{i=1}^{\rho}(-1)^{i+1}\omega_i - S_\theta(0).$$

We show the steps that lead to the above formulation. The wavelet filter $\psi(\lambda)$ consists of two components:

$$\psi(\lambda) = S_\theta(\lambda) + P_\omega(\lambda),$$

where $S_\theta(\lambda)$ and $P_\omega(\lambda)$ are the spectral and polynomial filter responses, respectively, at frequency $\lambda$. The admissibility condition requires that $\psi(0) = 0$, i.e.,

$$S_\theta(0) + P_\omega(0) = 0.$$

For the admissibility condition to hold, we either enforce both components to be zero ($S_\theta(0) = P_\omega(0) = 0$) or allow $P_\omega(0) = -S_\theta(0)$. We express $P_\omega(\lambda)$ as a Chebyshev polynomial. After rescaling the input from $\lambda \in [0,2]$ to $\widetilde{\lambda} \in [-1,1]$, the condition $P_\omega(0) = \widetilde{P}_\omega(-1)$ leads to the equation

$$\omega_0 = \sum_{i=1}^{\rho}(-1)^{i+1}\omega_i - S_\theta(0),$$

which satisfies the admissibility condition at zero frequency.

However, this approach is more challenging to implement, as it requires manually updating the learnable parameters. While this is feasible, we found that the primary approach worked better overall. It is more flexible and streamlined, making it easier to integrate into our model while still satisfying the necessary conditions.

### B.2.3 WAVELET RESIDUALS

We introduce residual connections within the wavelet filter (*wavelet residuals*) to improve model generalization. We write the forward step of our model with included wavelet residual connections as

$$\hat{\boldsymbol{H}}_{\psi,j}^{(l-1)} = \sigma\left[\left(\psi^{(l)}\left(\boldsymbol{U}, \boldsymbol{\Lambda}, \mathcal{L}\right) + \boldsymbol{I}\right)\hat{\boldsymbol{H}}^{(l-1)}\right] = \sigma\left[\psi_{\boldsymbol{I}}^{(l)}\left(\boldsymbol{U}, \boldsymbol{\Lambda}, \mathcal{L}\right)\hat{\boldsymbol{H}}^{(l-1)}\right]. \tag{14}$$

In the graph frequency domain, the forward step defined in the previous equation corresponds to the graph wavelet filter $\widehat{\psi}_{\boldsymbol{I}}(\boldsymbol{\Lambda}) = S_\theta(\boldsymbol{\Lambda}) + P_\omega(\boldsymbol{\Lambda}) + \boldsymbol{I}$. While including wavelet residual connections brings desirable properties in terms of gradient flow, preserving the theoretical properties of the wavelet transform remains important to theoretically guarantee the existence of the inverse wavelet transform. Similarly to how we enforced the admissibility condition in Appendix B.2.2, we can now let the spectral and polynomial filters transform as $S_\theta(\lambda) \to S_\theta(\lambda) - S_\theta(0) - 1/2$, and $P_\omega(\lambda) \to P_\omega(\lambda) - P_\omega(0) - 1/2$. In fact, this allows us to write $\widehat{\psi}_{\boldsymbol{I}}(\boldsymbol{\Lambda}) = S_\theta(\boldsymbol{\Lambda}) - S_\theta(\boldsymbol{0}) - \boldsymbol{I}/2 + P_\omega(\boldsymbol{\Lambda}) - P_\omega(\boldsymbol{0}) - \boldsymbol{I}/2 + \boldsymbol{I}$. The wavelet filter then becomes $\psi_{\boldsymbol{I}}\left(\boldsymbol{U}, \boldsymbol{\Lambda}, \mathcal{L}\right) = U\widetilde{S}_\theta(\boldsymbol{\Lambda})U^\top + \widetilde{P}_\omega(\mathcal{L}) + \boldsymbol{I}$, where $\widetilde{S}_\theta(\boldsymbol{\Lambda}) = S_\theta(\boldsymbol{\Lambda}) - S_\theta(\boldsymbol{0}) - \boldsymbol{I}/2$ and $\widetilde{P}_\omega(\mathcal{L}) = P_\omega(\mathcal{L}) - P_\omega(\boldsymbol{0}) - \boldsymbol{I}/2$, thus ensuring the condition $\widehat{\psi}(0) = 0$ by construction.

**Residual Connections.** While the inclusion of wavelet residuals theoretically ensures the preservation of the admissibility condition, in practice, we observe that normal residual connections, where the residual is added after the filtering operation, further improve the model's generalization. Therefore, we opt to use standard residual connections.

### B.3 COMPUTATIONAL COMPLEXITY

The LR-GWN layer integrates multiple structural components whose computational costs can be precisely characterized. Let $n$ and $m$ denote the number of nodes and edges in the input graph, and let $d$ represent the feature dimensionality, assumed constant across all layers of the network. We denote by $L$ the depth of the MLP $f_\vartheta^{(l)}$, by $\rho$ the polynomial order used in the Chebyshev approximation, by $k$ the number of retained eigenvectors for the spectral component, and by $J + 1$ the total number of filters, including the scaling function.

Each layer begins with a node-wise transformation via an $L$-layer MLP of hidden size $d$, shared across both filtering components. This projection incurs a cost of $\mathcal{O}(Ld^2n)$. The polynomial component uses recursive Chebyshev polynomials over the sparse Laplacian. Recall that multiplication between an $n \times n$ sparse matrix with $m$ non-zero entries, and a dense $n \times d$ matrix has cost $\mathcal{O}(md)$. For a $\rho$-order polynomial, and across all filters, this results in a total cost of $\mathcal{O}((J + 1)\rho md)$. The spectral component computes a projection via $U$ and back, for each filter. This involves two matrix multiplications of cost $\mathcal{O}(kdn)$ and an element-wise scaling of cost $\mathcal{O}(kd)$, which is asymptotically negligible. Across filters, the spectral component contributes a total cost of $\mathcal{O}((J + 1)kdn)$.

Crucially, the partial eigendecomposition required for the spectral component is computed once per graph at cost $\mathcal{O}(km)$ as a preprocessing step, and can be amortized across the network.

In summary, the per-layer complexity is $\mathcal{O}\left(Ld^2n + (J + 1)\rho dm + (J + 1)kdn\right)$. Since $J$, $L$, $d$, $\rho$, and $k$ are typically small constants in practice, the dominant terms grow with the number of nodes $n$, and edges $m$. The expression therefore simplifies to $\mathcal{O}(n + m)$. For sparse graphs we can assume constant node degree deg, hence $m = \deg \cdot n = \Omega(n)$, yielding an asymptotical complexity of $\mathcal{O}(m)$. This decomposition reveals the efficiency of LR-GWN: despite leveraging both spatial and spectral graph filters, the model remains scalable to large, sparse graphs.

## C EXPERIMENTAL DETAILS

Our codebase builds on the LRGB benchmark framework (Dwivedi et al., 2023)[1], as updated by Tönshoff et al. (2023)[2], which itself is based on the GraphGPS codebase (Rampášek et al., 2023)[3]. This foundation enabled fair and consistent model evaluation by providing a shared pipeline for data preprocessing, training, and testing. By reusing the same infrastructure, we ensured reliable comparisons and minimized implementation-related discrepancies. All experiments were run on Nvidia GTX 1080Ti GPUs with memory usage kept below 10GB.

### C.1 HYPERPARAMETER TUNING AND SELECTION

To optimize LR-GWN, we performed hyperparameter tuning using multiple sweeps to identify the best combination for each experiment. The hyperparameters were selected based on the highest validation performance, ensuring that the chosen configuration maximized the model's generalization ability. We explored a wide range of hyperparameters, including learning rate, batch size, number of layers, hidden dimension size, and wavelet-specific parameters. For each sweep, we conducted experiments across different values, evaluating the model's performance on the validation set after each configuration. Once the best-performing configuration was identified, it was used for the final testing phase. Table 4 summarizes the hyperparameters used for our experiments.

---

[1] https://github.com/vijaydwivedi75/lrgb
[2] https://github.com/toenshoff/LRGB
[3] https://github.com/rampasek/GraphGPS

Table 4: **Hyperparameters** of our model for experiments in Tables 1 and 2

|  | PEPTIDES-FUNC | PEPTIDES-STRUCT |
|---|---|---|
| PE/SE | MagLapPE (dim=16) | MagLapPE (dim=16) |
| Hidden dim. | 256 | 256 |
| Head depth | 2 | 2 |
| Wavelet layers | 2 | 2 |
| Num. wavelets | 10 | 10 |
| Spectral part | Yes | Yes |
| Spatial part | Yes | Yes |
| Admissibility | No | No |
| Gaussians | 60 | 60 |
| Cutoff $\lambda_{\max}$ | 0.9 | 0.9 |
| Polynomial order $\rho$ | 8 | 8 |
| Activation $\sigma$ | GeLU | GeLU |
| Dropout prob. | 0.25 | 0.25 |
| Skip connections | True | True |
| Batch size | 200 | 200 |
| Learning rate | 0.001 | 0.001 |
| Weight decay | 0.0 | 0.0001 |
| Epochs | 400 | 400 |
| Parameters | 248k | 248k |

## C.2 DATASETS

**Long-range interactions.** To evaluate the ability of models to capture long-range dependencies, we include the PEPTIDES-FUNC and PEPTIDES-STRUCT datasets from the Long-Range Graph Benchmark (LRGB) (Dwivedi et al., 2023; Tönshoff et al., 2023). These datasets are derived from SATPdb and are commonly used to assess how well models handle relationships extending beyond local neighborhoods. PEPTIDES-FUNC is a multi-label classification task with 10 peptide functional categories, while PEPTIDES-STRUCT is a regression task predicting 11 distinct 3D structural properties.

**Short-range interactions.** For short-range interactions, we consider the COMPUTERS and PHOTO datasets (Shchur et al., 2018), segments of the Amazon co-purchase graph (McAuley et al., 2015). In these graphs nodes represent goods, edges indicate that two goods are frequently bought together, node features are bag-of-words encoded product reviews, and class labels are given by the product category. These datasets primarily exhibit localized interactions, making them well-suited for evaluating traditional message-passing GNNs.

Table 5: **Dataset overview.** For each dataset, we list the source, license, propagation type, number of graphs, average number of nodes and edges, task, and evaluation metric.

| Dataset | Derived from | License | Propagation | # Graphs | Avg. # Nodes | Avg. # Edges | Task | Metric |
|---|---|---|---|---|---|---|---|---|
| PEPTIDES-FUNC (Dwivedi et al., 2022) | SATPdb | CC BY-NC 4.0 | Long-range | 15,535 | 150.9 | 307.3 | Graph classification | Avg. Precision |
| PEPTIDES-STRUCT (Dwivedi et al., 2022) | SATPdb | CC BY-NC 4.0 | Long-range | 15,535 | 150.9 | 307.3 | Graph regression | Mean Abs. Error |
| AMAZON PHOTO (Shchur et al., 2018) | McAuley et al. (2015) | Not provided | Short-range | 1 | 7,650 | 238,163 | Node classification | Accuracy |
| AMAZON COMPUTER (Shchur et al., 2018) | McAuley et al. (2015) | Not provided | Short-range | 1 | 13,752 | 491,722 | Node classification | Accuracy |

## D ADDITIONAL RESULTS

Beyond the wavelet-focused comparison presented in Section 5, we report extended results against a broader set of baselines in Tables 6 and 7. On long-range tasks (PEPTIDES-FUNC, PEPTIDES-STRUCT), LR-GWN consistently outperforms all prior wavelet-based models, including WaveGC (Liu et al., 2025), which was the strongest previous competitor on the LRGB benchmark. Among non-wavelet methods, LR-GWN achieves the best performance on PEPTIDES-FUNC, surpassing message-passing and transformer-style models such as GCN, PathNN, CIN++, and GPS. On PEPTIDES-STRUCT, where local features are especially predictive, GCN remains slightly ahead, but LR-GWN still matches or exceeds the majority of long-range baselines, highlighting its ability to capture both global and local structure.

Table 6: **Long-range benchmark results.** We report the mean and the unbiased standard error of the mean across multiple runs on PEPTIDES-FUNC and PEPTIDES-STRUCT. Results for baselines are either from (Tönshoff et al., 2023) or from respective papers (e.g., PathNN (Michel et al., 2023), CIN++ (Giusti et al., 2023), WaveGC (Liu et al., 2025)).

| | Method | PEPTIDES-FUNC ($\uparrow$) | PEPTIDES-STRUCT ($\downarrow$) |
|---|---|---|---|
| Others | GCN | $68.60 \pm 0.50$ | $\mathbf{24.60 \pm 0.07}$ |
| | GINE | $66.21 \pm 0.67$ | $24.73 \pm 0.17$ |
| | GatedGCN | $67.65 \pm 0.47$ | $24.77 \pm 0.09$ |
| | PathNN | $68.16 \pm 0.26$ | $25.45 \pm 0.32$ |
| | CIN++ | $65.69 \pm 1.17$ | $25.23 \pm 0.13$ |
| | GPS | $65.34 \pm 0.91$ | $25.09 \pm 0.14$ |
| Wavelet | SGWT | $60.23 \pm 0.27$ | $25.39 \pm 0.21$ |
| | GWNN | $65.47 \pm 0.48$ | $27.34 \pm 0.04$ |
| | DEFT | $66.95 \pm 0.63$ | $25.06 \pm 0.13$ |
| | WaveGC | $69.73 \pm 0.43$ | $24.83 \pm 0.11$ |
| | LR-GWN (**ours**) | $\mathbf{72.16 \pm 0.41}$ | $24.62 \pm 0.06$ |

Table 7: **Short-range benchmark results.** We report the mean and the unbiased standard error of the mean across multiple runs on PHOTO and COMPUTER. Results for baselines are either from Liu et al. (2025) or from respective papers (e.g., GWNN (Xu et al., 2019)).

| | Method | PHOTO ($\uparrow$) | COMPUTER ($\uparrow$) |
|---|---|---|---|
| Others | GCN | $92.70 \pm 0.20$ | $89.65 \pm 0.52$ |
| | GAT | $93.87 \pm 0.11$ | $90.78 \pm 0.13$ |
| | ChebNetII | $94.71 \pm 0.25$ | $89.85 \pm 0.85$ |
| Wavelet | SGWT | $92.45 \pm$ n.d. | $85.19 \pm$ n.d. |
| | GWNN | $94.45 \pm$ n.d. | $90.75 \pm$ n.d. |
| | ASWT-SGNN | $93.80 \pm$ n.d. | $89.40 \pm$ n.d. |
| | WaveGC | $\underline{95.37 \pm 0.44}$ | $\mathbf{92.26 \pm 0.18}$ |
| | LR-GWN (**ours**) | $\mathbf{95.69 \pm 0.23}$ | $\underline{91.15 \pm 0.08}$ |

On short-range tasks (PHOTO, COMPUTERS), LR-GWN achieves state-of-the-art accuracy on PHOTO and performs competitively on COMPUTERS, where WaveGC attains the top score. Compared to traditional message-passing GNNs such as GCN, GAT, and ChebNetII, LR-GWN consistently yields higher performance, demonstrating that the proposed hybrid design remains effective even when interactions are predominantly local. Taken together, these results confirm the conclusions from the main body: LR-GWN establishes a new state-of-the-art among wavelet-based approaches on long-range tasks, while remaining broadly competitive with non-wavelet baselines across both interaction scales.

# E THEORETICAL RESULTS

## E.1 POLYNOMIAL FILTERS IN SPECTRAL AND SPATIAL DOMAIN

For a graph with Laplacian $\mathcal{L} = \boldsymbol{U}\boldsymbol{\Lambda}\boldsymbol{U}^\top$ and a polynomial filter of degree $\rho$, the spectral filtering operation $\boldsymbol{U}\widehat{g}_\omega(\boldsymbol{\Lambda})\boldsymbol{U}^\top x$ is equivalent to the spatial operation $g_\omega(\mathcal{L})x$, where $\widehat{g}_\omega(\boldsymbol{\Lambda}) = \sum_{i=0}^{\rho} \omega_i \boldsymbol{\Lambda}^i$.

*Proof.* We prove this by showing that powers of the Laplacian in the spatial domain correspond to powers of eigenvalues in the spectral domain. Starting with the spectral filtering operation:

$$\boldsymbol{U} g_\omega(\boldsymbol{\Lambda}) \boldsymbol{U}^\top x = \boldsymbol{U} \left( \sum_{i=0}^{\rho} \omega_i \boldsymbol{\Lambda}^i \right) \boldsymbol{U}^\top x \tag{15}$$

$$= \sum_{i=0}^{\rho} \omega_i \boldsymbol{U} \boldsymbol{\Lambda}^i \boldsymbol{U}^\top x \tag{16}$$

Now we use the key insight that for the eigendecomposition $\mathcal{L} = \boldsymbol{U}\boldsymbol{\Lambda}\boldsymbol{U}^\top$, we have:

$$\boldsymbol{U}\boldsymbol{\Lambda}^i\boldsymbol{U}^\top = (\boldsymbol{U}\boldsymbol{\Lambda}\boldsymbol{U}^\top)^i = \mathcal{L}^i.$$

This holds by induction:

- Base case $i = 1$: $\boldsymbol{U}\boldsymbol{\Lambda}\boldsymbol{U}^\top = \mathcal{L}$ by definition

- Inductive step: Assume $\boldsymbol{U}\boldsymbol{\Lambda}^k\boldsymbol{U}^\top = \mathcal{L}^k$. Then:

$$\boldsymbol{U}\boldsymbol{\Lambda}^{k+1}\boldsymbol{U}^\top = \boldsymbol{U}\boldsymbol{\Lambda}^k\boldsymbol{\Lambda}\boldsymbol{U}^\top \tag{17}$$

$$= \boldsymbol{U}\boldsymbol{\Lambda}^k(\boldsymbol{U}^\top\boldsymbol{U})\boldsymbol{\Lambda}\boldsymbol{U}^\top \quad (\text{since } \boldsymbol{U}^\top\boldsymbol{U} = \boldsymbol{I}) \tag{18}$$

$$= (\boldsymbol{U}\boldsymbol{\Lambda}^k\boldsymbol{U}^\top)(\boldsymbol{U}\boldsymbol{\Lambda}\boldsymbol{U}^\top) \tag{19}$$

$$= \mathcal{L}^k\mathcal{L} = \mathcal{L}^{k+1} \tag{20}$$

Therefore:

$$\boldsymbol{U} g_\omega(\boldsymbol{\Lambda}) \boldsymbol{U}^\top x = \sum_{i=0}^{\rho} \omega_i \mathcal{L}^i x = g_\omega(\mathcal{L})x$$

This equivalence explains why polynomial spectral filters can be implemented efficiently without eigendecomposition—they correspond exactly to polynomial operations on the Laplacian matrix itself. $\qquad\square$

**Remark.** This result extends immediately to any polynomial basis (Chebyshev, Bernstein, etc.) since they are all linear combinations of monomials.

E.2    PROOF OF LEMMA 1

*Proof.* We begin by expanding the left-hand side of the expression. Distributing the product over the sum gives

$$\boldsymbol{U} S(\boldsymbol{\Lambda}) \boldsymbol{U}^\top + \boldsymbol{U} P(\boldsymbol{\Lambda}) \boldsymbol{U}^\top. \tag{21}$$

Next, we focus on the term $P(\boldsymbol{\Lambda})$, which is parameterized as a polynomial function of the eigenvalues of the graph Laplacian $\boldsymbol{\Lambda}$. Specifically, $P(\boldsymbol{\Lambda})$ can be written as

$$P(\boldsymbol{\Lambda}) = \sum_{i=0}^{\rho} \gamma_i \boldsymbol{\Lambda}^i, \tag{22}$$

where $\gamma_i$ are learnable coefficients and $\rho$ is the polynomial degree. Substituting this expansion for $P(\boldsymbol{\Lambda})$ into the expression, we get:

$$\boldsymbol{U} P(\boldsymbol{\Lambda}) \boldsymbol{U}^\top = \boldsymbol{U} \left( \sum_{i=0}^{\rho} \gamma_i \boldsymbol{\Lambda}^i \right) \boldsymbol{U}^\top \tag{23}$$

$$= \sum_{i=0}^{\rho} \gamma_i \boldsymbol{U} \boldsymbol{\Lambda}^i \boldsymbol{U}^\top \tag{24}$$

The key insight is that for the eigendecomposition $\mathcal{L} = \boldsymbol{U}\boldsymbol{\Lambda}\boldsymbol{U}^\top$, we have:

$$\boldsymbol{U}\boldsymbol{\Lambda}^i\boldsymbol{U}^\top = \mathcal{L}^i. \tag{25}$$

This holds by induction as shown in Appendix E.1. Substituting Eq. (25) into Eq. (24), we obtain

$$\boldsymbol{U}P(\boldsymbol{\Lambda})\boldsymbol{U}^\top = \sum_{i=0}^{\rho} \gamma_i \mathcal{L}^i = P(\mathcal{L}). \tag{26}$$

Finally, combining the two terms from Eq. (21), we get the desired decomposition

$$\boldsymbol{U}\left[S(\boldsymbol{\Lambda}) + P(\boldsymbol{\Lambda})\right]\boldsymbol{U}^\top = \boldsymbol{U}S(\boldsymbol{\Lambda})\boldsymbol{U}^\top + P(\mathcal{L}). \tag{27}$$

This completes the proof. $\qquad\square$

