# OpenReview forum: "Long-Range Graph Wavelet Networks"
_ICLR.cc/2026/Conference — ICLR 2026 Conference Withdrawn Submission_

### Official Review · Reviewer_Asoa · 2025-10-27

**Soundness:** 2
**Presentation:** 3
**Contribution:** 2
**Rating:** 2
**Confidence:** 4

**Summary:**

This paper proposed a new graph wavelet, a polynomial backbone plus a spectral correction. This technique resolved the locality bottleneck, preserved the interpretability, and achieved linear complexity with O(E). Experiments showed the improvements over baselines.

**Strengths:**

1. The proposed wavelet utilized a learnable spectral function to mitigate the local propagation.
2. This paper is theoretically supported.
3. The organization is good and clear to follow.

**Weaknesses:**

1. For a wavelet, scales are the ones to control the receptive fields. However, the authors did not mention this very important wavelet component in their model design.
2. In section 3, the authors listed three theorems. But these theorems seem not relevant to the proposed model. If we skip this section, we can still understand other details.
3. One of the claimed strengths is the linear complexity. This linear complexity is just because of advanced EVD algorithm, rather than special model design. A potential evidence is the report of running time. Actually, EVD part took very short time, but LR-GWN took more time than that reported in WaveGC.
4. The authors took 4 datasets, two long-range and two short-range. It's not enough to verify the generalization of the model.

**Questions:**

How to understand so-called "interpretability of pure wavelet  propagation" at the begining of section 4? In the paper, I do not feel where the interpretatbility is. What's its benefit?

---

### Official Review · Reviewer_R4kQ · 2025-11-01

**Soundness:** 2
**Presentation:** 3
**Contribution:** 2
**Rating:** 2
**Confidence:** 5

**Summary:**

This paper proposes a new type of long range graph wavelet by combining the polynomial filters and the spectral filters with admissibility guaranteed. Compared to traditional spectral filters based on eigenvalue decomposition (EVD), the proposed graph wavelet achieves long range perception with higher efficiency. Long-Range Graph Wavelet Networks (LR-GWNs) are then developed and achieve state-of-the-art performance in node classification and node regression.

**Strengths:**

1. The long range graph wavelet developed in this paper achieves a global perception field with much lower computational complexity than wavelets constructed via EVD.

2. Graph wavelet networks based on wavelet decomposition are developed to achieve multi-resolution representation learning. LR-GWN is applicable for both long range and short range tasks.

**Weaknesses:**

1. The methodology to construct the long range wavelet is of limited novelty. This paper mixes two fundamental approaches of polynomial filters based on graph shift operators (used in ChebNet (Defferrard et al., 2016)) and graph wavelets based on EVD (used in GWNN (Xu et al. 2019)). These two methods are commonly used in developing spectral graph convolution, and the long-range wavelet in this paper is equivalent to a combination of the two methods. In fact, it is realized using a set of polynomial filters (corresponding to ChebNet) and wavelet filters based on the first k low-frequency components (corresponding to GWNN).

2. The claim on the proposed wavelet that all "wavelets" contain low frequency components is confusing. When all the filters could contain the same frequency components, the wavelets could be no longer tightly supported in the spatial domain and cause redundancy in multi-resolution representation learning. This contradicts to compact support in the spatial and frequency domains, which is one of the most important property of wavelets. In fact, the long-range wavelet may not actually be a wavelet, since the term $S_{\theta_\psi^{(l)}}(\Lambda)$ seems to contain only the first k frequency components and violate the spatial compactness of wavelets.

3.  Theoretical analysis on frame bounds for the proposed wavelet is missing. Considering that this paper does not naturally inherit the theoretical merits of wavelets using well-established graph wavelets like (Xu et al. 2019), it cannot establish the theoretical properties of redundancy in representation and stability of the proposed graph wavelet networks like other works [R1].

4. Experiments are not sufficient to show the LR-GWN has universal advantages in performance compared with other methods since commonly used datasets (such as CORA, CITESEER, PUBMED for node classification and ZINC for node regression) are not included.

[R1] A. Tong et al., "Learnable Filters for Geometric Scattering Modules," in IEEE Transactions on Signal Processing, vol. 72, pp. 2939-2952, 2024, doi: 10.1109/TSP.2024.3378001.

**Questions:**

Please refer to the section of Weaknesses.

---

### Official Review · Reviewer_ZrXG · 2025-11-11

**Soundness:** 3
**Presentation:** 4
**Contribution:** 2
**Rating:** 6
**Confidence:** 3

**Summary:**

* Motivation: Polynomial graph filters avoid eigendecomposition and are efficient, but a degree-$\rho$ filter only mixes information within $\rho$-hops and lacks the resolution needed for low-frequency/long-range control. Capturing long-range behaviour typically requires large $\rho$, which increases costs.
* Core idea: Augment a polynomial filter with a low-frequency spectral component computed via a partial EVD, to regain control over low frequencies (long-range dependencies) without a full decomposition.
* Each wavelet kernel is decomposed into a polynomial part and a low-frequency spectral part:
    \begin{align}
        \kappa * x = U[P(\Lambda) + S(\Lambda)]U^\top x.
    \end{align}
    $P(\Lambda)$ is a finite order (e.g. Chebyshev) polynomial with parameter $\omega$, and $S(\Lambda)$ is a low-frequency controller with parameter $\theta$.  The polynomial term provides local mixing, whereas the spectral term provides low-frequency control for long-range effects.
* To define the low-pass eigenspace, they set a cutoff $\lambda_{cut} < \lambda_{max}$ and selects eigenpairs (eigenvalue $\lambda_i$ and its corresponding eigevector $u_i$) with $\lambda_i \leq \lambda_{cut}$.  Determining an ideal $\lambda_{cut}$ would, in principle,require a full eigendecomposition.  In practice, they fix a $k$ and computes only the $k$ smallest eigenpairs using a Lanczos algorithm, once per graph as preprocessing, thereby avoiding the cubic cost of a full EVD.
* Low-pass spectral components then operates only in the subspace spanned by these $k$ eigenvectors (corresponding to the $k$ smallest eigenvalues), so the computation is $U_k S_{\theta}(\Lambda_k) U_k^{\top} x$. Thus, $k$ caps the dimensionality of the low-frequency space per layer.

**Strengths:**

The paper is well written. The exposition of the motivation and the proposed methods is clear. The authors explicitly discuss limitations, which I appreciate as a fair and balanced critique.

**Weaknesses:**

* There is a clear trade-off. On some problems, using only the lowest $k$ eigenpairs may be insufficient.  If $k$ is too small, the model may still lack long-range/low-frequency control. If $k$ is too large, compute and memory are wasted. Because both the per-layer cost and the partial EVD preprocessing scale linearly with $k$, a large $k$ can become comparable to a full EVD and negate the efficiency gains of the polynomial approximation. A more principled sliding-scale mechanism—learning when to downplay the polynomial path versus rely more on the low-frequency partial EVD, may be of help.  However, this is beyond the scope of the current algorithm. As presented, the hardness of the problem relies entirely on tuning $k$.
* Because the method relies on a partial EVD, the top $k$ eigenspace $U\in\mathbb{R}^{n\times k}$ must be computed and stored. This requires access to all $n$ nodes and edges, and when $n$ is large, both memory use and computation grow linearly with $n$. By contrast, purely polynomial GNNs avoid this exact problem of $n$ factor in their computational complexity, albeit with some loss in accuracy. Introducing a partial EVD therefore reintroduces the same concern that motivated the use of polynomial approximations in the first place.

**Questions:**

See Weakness section

---

### Official Review · Reviewer_C9m8 · 2025-11-11

**Soundness:** 2
**Presentation:** 2
**Contribution:** 2
**Rating:** 2
**Confidence:** 5

**Summary:**

This paper proposes long-range graph wavelet networks (LR-GWN) that construct wavelet filters using a combination of complementary low-order local polynomial and global spectral correction. The hybrid scheme contains low-order polynomials are efficient to achieve local aggregation and spectral correction is realized to capture long-range interactions to unify short- and long-distance information flow within a principled wavelet framework. Experiments show that LR-GWN outperforms compared wavelet-based methods on long-range benchmarks and is competitive on short-range datasets.

**Strengths:**

1. This paper introduces an interesting idea of long-range graph wavelets

2. The proposed wavelets leverage polynomial filter and spectral correction to capture local and global interactions.

**Weaknesses:**

1. This paper proposes hybrid parameterization of wavelet filters and principled wavelet propagations, but the rationale and principle for this construction are not well clarified.

i) The motivation for this formulation is vague. The constructed wavelets are not shown to enjoy the properties of classical wavelets such as guaranteeing stable reconstruction, reducing overlapping between multiple bands, and achieving spatial-spectral localization. Under such consideration, the benefits of constructing “wavelets” rather than directly learning a sequence of graph filters with varying frequence response are not clear.

ii) The scaling function and wavelet function are supposed to capture low-frequency and high-frequency components in principle. However, in the paper, both functions are formulated using polynomial functions and spectral correction, and constraints to force the two functions to achieve low-pass and high-pass filtering are not mentioned. The difference between the scaling function and wavelet function in the paper and whether they are corresponding to low-pass and high-pass filtering are not clear.

2. The proposed wavelets are in fact a combination of a polynomial filter and a spectral filter. This idea of employing a sequence of multiscale filters has been similarly explored in graph scattering networks (e.g., [7]-[13]) that are usually an extension of wavelets. However, these works are not discussed and compared in the paper.

3. Experimental evaluations are not sufficient to support the claims of state-of-the-art performance and generalizability.

i) Comparison with well-known graph transformer architectures that are specifically designed to handle long-range dependencies and mitigate over-squashing such as Graphormer [1], SAN [2], GraphIT [3] is missing. These models have been widely adopted for long-range graph benchmarks.

ii) Comparison with wavelet-based GNNs is limited. Classical or recent spectral GNNs and wavelet-based GNNs such as BernNet [4], ChebNetII [5], and Specformer [6] and LEGS [7] are not evaluated and discussed in the paper.

iii) This paper claims to mitigates the computation prohibitive full eigenvalue decomposition of graph Laplacians, but does not provide experimental results on computation time and parameter complexity for validation.

[1] Y. Shi et al. Benchmarking graphormer on large-scale molecular modeling datasets. arXiv preprint arXiv:2203.04810, 2022.

[2] D. Kreuzer, D. Beaini, W. L. Hamilton, V. Létourneau, and P. Tossou. Rethinking graph transformers with spectral attention. NeurIPS 2021.

[3] G. Mialon, D. Chen, M. Selosse, J. Mairal. Graphit: Encoding graph structure in transformers. arXiv preprint arXiv:2106.05667, 2021.

[4] He M, Wei Z, Xu H. Bernnet: Learning arbitrary graph spectral filters via Bernstein approximation. NeurIPS 2021.

[5] He M, Wei Z, Wen J R. Convolutional neural networks on graphs with chebyshev approximation, revisited. NeurIPS 2022.

[6] D. Bo, C. Shi, L. Wang, and R. Liao. Specformer: Spectral graph neural networks meet Transformers. ICLR 2023.

[7] A. Tong et al. Learnable filters for geometric scattering modules. IEEE Transactions on Signal Processing. 2024, 72: 2939-2952.

[8] Y. Min, F. Wenkel, and G. Wolf. Scattering GCN: Overcoming oversmoothness in graph convolutional networks. NeurIPS 2020.

[9] F. Gama, J. Bruna, and A. Ribeiro. Stability of graph scattering transforms. NeurIPS2019.

[10] D. Zou and G. Lerman. Graph convolutional neural networks via scattering. Applied and Computational Harmonic Analysis, vol. 49, no. 3, pp. 1046–1074, 2020.

[11] F. Gao, G. Wolf, and M. Hirn. Geometric scattering for graph data analysis. ICML 2019, pp. 2122–2131.

[12] J. Chew et al. Geometric scattering on measure spaces. Applied and Computational Harmonic Analysis, 2024, 70: 101635.

[13] C. Koke and G. Kutyniok, “Graph scattering beyond wavelet shackles,” NeurIPS 2022.

**Questions:**

Please refer to the Weaknesses.

---

### Note · Authors · 2025-11-20

**Comment:**

Dear Area Chairs and Reviewers,

Thank you for the thorough feedback on our submission.
After careful consideration of your comments, we have decided to withdraw the paper from consideration at ICLR 2026 so that we can revise and strengthen it, particularly by addressing the points you raised and integrating them with greater detail and care.

We appreciate the time and effort you invested in reviewing our work.

**Withdrawal Confirmation:**

I have read and agree with the venue's withdrawal policy on behalf of myself and my co-authors.